# Real-Time 3D Imaging and Inhibition Analysis of Human Serum Amyloid A Aggregations Using Quantum Dots

**DOI:** 10.3390/ijms252011128

**Published:** 2024-10-16

**Authors:** Liangquan Shi, Gegentuya Huanood, Shuto Miura, Masahiro Kuragano, Kiyotaka Tokuraku

**Affiliations:** Graduate School of Engineering, Muroran Institute of Technology, Muroran 050-8585, Japan; shi199510262022@163.com (L.S.); gegentuya1996@gmail.com (G.H.); 23041072@muroran-it.ac.jp (S.M.); gano@muroran-it.ac.jp (M.K.)

**Keywords:** quantum dot nanoprobes, rosmarinic acid, serum amyloid A

## Abstract

Serum amyloid A (SAA) is one of the most important precursor amyloid proteins discovered during the study of amyloidosis, but its underlying aggregation mechanism has not yet been well elucidated. Since SAA aggregation is a key step in the pathogenesis of AA amyloidosis, amyloid inhibitors can be used as a tool to study its pathogenesis. Previously, we reported a novel microliter-scale high-throughput screening (MSHTS) system for screening amyloid β (Aβ) aggregation inhibitors based on quantum dot (QD) fluorescence imaging technology. In this study, we report the aggregation of human SAA (hSAA) in phosphate-buffered saline, in which we successfully visualized hSAA aggregation by QD using fluorescence microscopy and confocal microscopy. Two-dimensional and three-dimensional image analyses showed that most aggregations were observed at 40 μM hSAA, which was the optimal aggregation concentration in vitro. The accuracy of this finding was verified by a Thioflavin T assay. The transmission electron microscopy results showed that QD uniformly bound to hSAA aggregation. hSAA aggregation inhibitory activity was also evaluated by rosmarinic acid (RA). The results showed that RA, which is a compound with high inhibitory activity against Aβ aggregation, also exhibited high inhibitory activity against 40 μM hSAA. These results indicate that the MSHTS system is an effective tool for visualizing hSAA aggregation and for screening highly active inhibitors.

## 1. Introduction

Amyloidosis is a group of clinical syndromes caused by various factors [1,2]. It is not a clinically independent disease; rather, abnormal proteins form fibrillar tissue deposits that can damage key internal organs and lead to early death [3,4]. So far, more than 50 proteins or short peptides have been extracted from tissues of patients with various diseases, and their corresponding chemical components have been further analyzed and understood, which has significantly improved the rate of diagnosis [5,6]. Amyloidosis is not uncommon in clinical practice. It is characterized by the deposition of amyloid protein-related substances in blood vessel walls and tissues, causing lesions, and mainly accumulating in organs and tissues such as the heart, kidneys, liver, spleen, gastrointestinal tract, joint muscles, and skin [7].

AA amyloidosis, also known as secondary amyloidosis, is the most common systemic form of amyloidosis. Its causes are mainly long-term infectious (such as osteomyelitis, tuberculosis) or non-infectious (such as rheumatoid arthritis, familial Mediterranean fever) inflammation and a few tumors (such as hepatic adenoma, renal cell carcinoma, mesothelioma) [8,9,10]. In AA amyloidosis, the amyloid fibrils are composed of serum amyloid A (SAA), which belongs to a family of very closely related and highly conserved serum molecules [11]. SAA has been found to play an important role in lipid metabolism, to contribute to bacterial clearance, inflammation regulation, and antiviral activities [12], and to be a major serum acute phase protein [13]. When the body is infected or injured, most SAA is synthesized in the liver under the control of cytokines (especially IL-1, IL-6, and TNF) [14,15]. However, it has also been reported to be synthesized in small amounts in organs such as the intestines, breast, lungs, and uterus [16,17]. Therefore, it is also used as an inflammation marker in clinical practice [18]. There is a small amount of SAA in the healthy human body, but when the body is in an inflammatory state, its concentration can rapidly increase by about 1000 times within 4–6 h [19]. In certain chronic inflammatory states, high SAA concentrations in plasma can persist for several months, causing the C-terminal region of SAA to be easily cleaved, misfolded, and aggregated into an insoluble β-sheet form, leading to AA amyloidosis [20]. Therefore, a high SAA concentration in the blood is considered a risk factor for the development of AA amyloidosis. Although a variety of drugs have been considered, there is currently no established treatment for AA amyloidosis, and inhibition of SAA levels remains the primary goal of treatment [21]. Therefore, current research goals should focus on screening highly active inhibitors.

Previously, we reported a new microliter-scale high-throughput screening (MSHTS) system for screening Aβ aggregation inhibitors based on quantum dot (QD) fluorescence imaging technology [22]. QDs are characterized by their long-term photo-stability, chemical and physical stability, nanoscale size, and multi-color fluorescence emission from a single excitation, capabilities that are useful for long-term single-molecule imaging in vitro and in vivo [23]. With improvements in the system, it has now become possible to achieve high-throughput, automated, rapid, and accurate analysis [24]. The MSHTS screening system can estimate the half-maximal effective concentration (EC_50_) of inhibitory activity by analyzing only 5 μL of sample volume and has been shown to be accurate for crude extracts containing a wide range of contaminants [25]. We also used this system to screen out high Aβ aggregation inhibitory compounds from 52 ethanol spice extracts, the main component of which was rosmarinic acid (RA) [22]. RA is known to exhibit a variety of pharmacological effects, such as heart, liver, and lung protection, as well as antioxidant and anti-inflammatory effects [26].

Most studies on SAA aggregation have described amyloid formation by various protein isoforms, but not much research has been conducted on human SAA (hSAA) aggregation. In this study, we successfully observed the aggregation of hSAA based on the QD-based MSHTS system by using Two-dimensional (2D) and three-dimensional (3D) fluorescence microscopy (Figure 1). By using transmission electron microscopy (TEM), we detected an amyloid fibril structure after hSAA was incubated with QDs. The inhibitory activity of different concentrations of RA on hSAA aggregation was evaluated. The results showed that RA inhibited hSAA aggregation, and low concentrations of RA (2.4 μM and 12 μM) were better at inhibiting hSAA aggregation than Aβ aggregation. These results demonstrate that the MSHTS system can provide a new basis for clarifying the mechanism of amyloid protein formation while serving as a useful technology for screening inhibitors.

## 2. Results

### 2.1. Imaging hSAA Aggregation Using QDs

In previous studies, we demonstrated that QDs can be used to observe recombinant mouse SAA (mSAA) aggregation under fluorescence microscopy and quantified the amount of aggregates from microscopic images [27]. Here, we used recombinant hSAA to obtain images of amyloid aggregation using QDs [5,22]. We estimated the amount of amyloid aggregation from the standard deviation (SD) value (variation in the brightness of each pixel in the image) in fluorescence microscopic images according to the methodology employed in previous studies [5,22,23,24,25]. Aggregation was observed after 24 h of incubation, increasing over time (Figure 2A). We used ImageJ software (National Institutes of Health) to analyze 2D images of hSAA to determine SD values and found that 40 μM hSAA had the highest SD value (Figure 2B). The aggregation of 40 μM hSAA stabilized after 144 h of incubation, while other concentrations stabilized after 96 h of incubation (Figure 2B).

### 2.2. Three-Dimensional Observation of hSAA Aggregation

To gain insight into hSAA aggregation, we used confocal microscopy to observe 3D images of hSAA cultured for 168 h. The shape of hSAA aggregation could be visualized by using confocal laser microscopy (Figure 3). The hSAA aggregation at different concentrations in 3D imaging increased over time, with the largest aggregation at 40 μM hSAA, which was consistent with the results of the 2D images. To understand the hSAA aggregation structure and generation process more intuitively and stereoscopically, we rotated the three-dimensional 40 μM hSAA aggregation image 360° clockwise from top to bottom. Those results showed that aggregation was distributed in a dotted manner (Appendix A). Then, a dynamic video was generated based on the three-dimensional 40 μM hSAA aggregation image and incubation time, which was characterized by the continuous thickening of the aggregates over time, forming larger irregular aggregation (Appendix A). Based on the results of Figure 2 and Figure 3, the following experiments were performed in 40 μM hSAA solution.

### 2.3. Thioflavin T (ThT) Fluorescence Measurement Results of hSAA Protein Aggregation

As a common method for amyloid quantification, we incubated hSAA with ThT and measured its fluorescence intensity in real time. After 24 h of observation, we statistically analyzed the ThT fluorescence of 40 μM hSAA. By comparing the fluorescence intensity after 24 h of incubation, we found that the fluorescence intensity of the hSAA group was significantly higher than that of the two other control groups (Figure 4A). This result demonstrates that 40 μM hSAA formed amyloid fibrils after 24 h of incubation.

### 2.4. Visualization of hSAA Aggregation by TEM

TEM was used to determine whether hSAA formed fibrils and also to assess whether the QDs were bound to the fibrils. As shown in the images, hSAA had misfolded fibrils in vitro, which exhibited a linear morphology and irregular length, and greater hSAA aggregation formed a loose mesh structure (Figure 4B,C). After incubation of hSAA with QDs at 37 °C for 24 h, the fibrils were uniformly bound to the QDs (Figure 4B). The hSAA aggregation formation was independent of QDs (Figure 4C).

### 2.5. Effect of RA on hSAA Aggregation

Since RA samples need to be dissolved with ethanol (EtOH), it was first necessary to evaluate the effect of adding different concentrations of EtOH to PBS on hSAA aggregation. After incubation for 24 h, hSAA aggregation was observed, but different concentrations of EtOH had different effects on hSAA aggregation (Figure 5A). ImageJ software was used to analyze the 2D images to determine the SD values. By analyzing the SD values at 0 h and 24 h, we found that 2.5% EtOH inhibited hSAA aggregation, while 5%, 10%, and 20% EtOH significantly promoted hSAA aggregation (Figure 5B). Since 2.5% EtOH inhibits hSAA aggregation, it will affect the evaluation results of RA, while 10% and 20% EtOH may have toxic effects on subsequent planned cell experiments. Therefore, we selected 5% EtOH to add to PBS to evaluate the inhibitory activity of RA on hSAA aggregation. The hSAA and Aβ were dissolved in DMSO and diluted with PBS and 5% EtOH. The final DMSO concentrations were 9.6% and 2.5%, respectively. Previously, studies have shown that DMSO concentrations below 10% do not affect the inhibition results [24]. The results show that RA had an inhibitory effect on the aggregation of hSAA and Aβ in vitro. After 24 h of incubation, 2.4 μM and 12 μM RA had no significant inhibitory effect on Aβ aggregation, while 60 μM RA showed a significant inhibitory effect. For hSAA aggregation, 2.4 μM and 12 μM RA showed an inhibitory effect (Figure 5C). By comparing the SD values of the inhibition of RA on Aβ and hSAA aggregation in the 2.4 μM and 12 μM RA treatments, the aggregation inhibitory activity of RA on hSAA was shown to be significantly higher than that of Aβ (Figure 5D).

## 3. Discussion

SAA is one of the most important precursor amyloid proteins. Despite breakthroughs in the structure and molecular mechanism of SAA, its underlying aggregation process and pathogenic mechanism remain largely unknown [28]. For years, direct targeted treatments to remove amyloid aggregation and deposition in organs or tissues were not successful, and existing treatments have not been able to reverse or prevent the progression of this disease [29,30].

In this study, we used the MSHTS system and fluorescence and confocal microscopy imaging to elucidate the hSAA aggregation process and evaluate the effect of RA as an inhibitor. The QDs in the experiment were purchased from Thermo Fisher Scientific and were not labeled. Therefore, after hSAA was incubated with QDs in a 1536-well plate at 37 °C, the QDs bound nonspecifically to hSAA aggregation, allowing the images of the aggregates to be captured by fluorescence microscopy. The aggregation of hSAA at different concentrations was observed over time by confocal and fluorescence microscopy (Figure 2A and Figure 3). Previously, we used this method to elucidate the aggregation process of various amyloid proteins (tau, α-synuclein proteins) and mSAA [5]. The above descriptions of the imaging of the aggregation process are all based on the nonspecific binding between the QDs and protein aggregates. The amino groups of the QDs that we used repel each other, so the QDs do not aggregate by themselves. In fact, no aggregation was observed in the 0 μM hSAA samples in Figure 2A and Figure 3. This clearly indicates that the aggregation observed here is not dependent on the properties of the QDs. Although the mechanism by which QDs bind to amyloid aggregates such as SAA and Aβ remains unclear, TEM observations have confirmed that QDs indeed bind to the aggregations [5]. Therefore, this method is only suitable for observing one type of amyloid aggregation in vitro. To solve these problems, we coupled QDs with Aβ to form a stable QD-Aβ fluorescent probe.

The outer surfaces of QDs were covered with a polyethylene glycol (PEG) coating to cover the core surface cadmium atoms, which greatly reduced the cytotoxicity and achieved in vivo microglia, co-localization of lysosomes, and detection of amyloid protein [23,31,32,33]. Although PEG coating did not display cytotoxicity, we still replaced it with carbon QDs [34]. The carbon QDs are mainly composed of carbon, hydrogen, oxygen, and other elements and do not contain toxic heavy metal elements; they have good biocompatibility and can exist stably in organisms [35]. The carbon QDs can be used as a high-performance and non-toxic fluorescent probe for single-photon and multi-photon bio-imaging in vitro and in vivo, competing not only with traditional molecular dyes but also with traditional semiconductor QDs [35]. Next, we will consider the coupling of QDs of different sizes with other amyloid proteins and precursor proteins. We want to explore whether different fluorescence colors emitted by QDs of different sizes will be able to successfully elucidate the aggregation of multiple amyloid proteins and precursor proteins in vivo and identify different protein types.

The SD values were analyzed and plotted into a typical kinetic curve of hSAA aggregation, which also verified that the SD values increased over time (Figure 2B). We previously reported that 50 μM mSAA was the optimal concentration for quantifying inhibitory activity [27]. The results in this study showed that 40 μM hSAA, which showed the maximum SD value, was the optimal concentration for quantifying inhibitory activity (Figure 2B). We compared these results and found that under the same concentration fluorescence images, hSAA aggregations were less than mSAA aggregations. This result was also confirmed by the SD value. To appreciate whether the in vitro aggregation of mice and humans is similar or the same, in the future, we will use mice as an experimental model to study the in vivo aggregation of mSAA and the in vivo evaluation of inhibitors to provide a reference for in vivo studies of hSAA. Aβ and mSAA began to aggregate at around 5 h and 2 h, and the aggregation of Aβ and mSAA rate reached saturation within 24 h and 48 h [27,36]. However, hSAA aggregation occurred later than Aβ and mSAA, and its aggregation rate was much lower than that of Aβ and mSAA. This means that in clinical practice, the process of hSAA misfolding and aggregation requires much more time than the aggregation of Aβ and mSAA. If it is possible to detect or take preventive measures at an early stage of hSAA aggregation, this will allow for treatment time in clinical practice to be considerably extended. We also analyzed the structural details of hSAA aggregation by confocal microscopy. The hSAA 3D aggregation structure is very similar to the amyloid proteins we reported previously, which aggregate in a helical manner and exhibit a mesh-like structure [5]. The TEM results showed that the fibrils were uniformly bound to the QDs. These results suggest that imaging 3D aggregation by QDs may help to understand the 3D structure of amyloid proteins and reveal the mechanism of amyloid aggregation.

To verify the accuracy of the MSHTS experimental results, we performed a ThT experiment on hSAA aggregation because amyloid fibrils can specifically bind to ThT, absorb fluorescence at around 445 nm, and emit very high fluorescence at around 485 nm. The ThT results were consistent with the MSHTS results, and both results prove the in vitro aggregation of hSAA. The fluorescence signal of ThT is very sensitive to charge, environmental conditions, and complex samples, which may lead to false positive results [37,38]. The MSHTS system has now made high-throughput, automated, rapid, and accurate analyses possible, and it is accurate for crude extracts containing a variety of contaminants [25]. The MSHTS system will be advantageous in research fields pertaining to amyloid aggregation and inhibitor screening.

The method of synthesizing drugs based on small peptides selected from amyloid protein sequences has been successfully applied to inhibit or slow down the aggregation of Aβ (connected to Alzheimer’s disease) and α-synuclein (implied in Parkinson’s disease) and SAA [39,40,41]. However, one of our research objectives is to inhibit the deposition of SAA in organs using natural plant extracts. RA, a polyphenol with high Aβ inhibitory activity, was screened from 52 spices using the MSHTS system [22]. Moreover, by adding 1% RA to a mouse diet, it effectively inhibited the deposition of amyloid in organs and also improved the inhibitory effect of mouse serum on mSAA aggregation [27]. This is consistent with the positive results of the aggregation inhibition of hSAA in vitro in this study. At present, the mechanism by which RA and monoamine inhibit Aβ aggregation is through the o-quinone structure [42]. Studies have shown that the o-quinone structure specifically binds to Aβ and inhibits the further aggregation of Aβ and oligomers [43]. It has also been reported that compounds with an o-quinone structure covalently bind to nucleophilic amino acid residues to inhibit the extension of Aβ fibrils [44]. We speculate that the structure of RA itself reacts chemically with hSAA, making hSAA more stable, thus avoiding β-sheet misfolding.

In this study, we elucidated the in vitro hSAA aggregation characteristics using the MSHTS system of unlabeled QDs, which can detect the fluorescence intensity of a total of 1536 samples at a time, requiring only 5 µL per sample. Moreover, the system can be automated and used to accurately and precisely evaluate the inhibitory activity, even for crude extracts of natural products and commercial dressings [24,25].

In summary, after years of optimization, our independently developed MSHTS system has shown potential value in amyloid aggregation and screening inhibitors, although the application of this system is not limited to these aspects alone. In the future, not only will our research results help to better understand the pathogenesis of AA amyloidosis but we also aim to evaluate the inhibitors screened by this system with a wider range of in vitro and in vivo experiments and develop them into functional foods for the prevention of AA amyloidosis.

## 4. Materials and Methods

### 4.1. Materials

Recombinant human Serum Amyloid A Protein (Active) (No. ab285550), Human Amyloid Peptide of Aβ (4349-v), and Qdot™ 605 ITK™ Amino (PEG) Quantum Dots (Q21501MP) were purchased from Abcam (Cambridge, UK), Peptide Institute Inc. (Osaka, Japan), and Thermo Fisher Scientific (Waltham, MA, USA), respectively.

### 4.2. Imaging of hSAA Aggregation by QDs

Recombinant hSAA samples for fluorescence microscopic observation were prepared as follows: 0, 5, 10, 20, 30, 40, and 50 µM hSAA and 30 nM QD_605_ in PBS. A total of 5 μL of sample was transferred to each well of a 1536-well plate (782096, Greiner, Kremsmünster, Austria) and the plate was centrifuged at 3700 rpm for 5 min. After centrifugation, the plates were placed in a thermostatic incubator (SIB-35, Sansyo, Tokyo, Japan) and incubated at 37 °C. Images were observed and recorded every 24 h using an inverted fluorescence microscope (TE2000, Nikon, Tokyo, Japan) equipped with a color CCD camera (DP72, Olympus, Tokyo, Japan) or a confocal laser microscope system (Nikon C2 Plus, Nikon). To estimate the amount of aggregation from 2D images, the SD values of the fluorescent intensity of 200 × 200 pixels in the central region of fluorescence graphs were measured by ImageJ software version 1.53b (National Institutes of Health, Bethesda, MD, USA).

### 4.3. ThT Assay

A total of 40 μM ThT was mixed with 80 μM hSAA, 50 μM Aβ, DMSO, and PBS separately into tubes. The final concentrations of hSAA, Aβ, and ThT were 40 μM, 25 μM, and 20 μM, respectively. In total, 100 μL of sample was injected into a 96-well plate (5866-096, Iwaki Agc Techno Glass Co., Ltd. Tokyo, Japan), sealed, and centrifuged briefly. The ThT fluorescence signal was measured by a microplate reader (SH-9000, Yamato, Tokyo, Japan) with the following settings: 450 nm excitation wavelength; 490 nm emission wavelength.

### 4.4. Observation of hSAA Aggregation by TEM

The total volume of 40 μM hSAA samples was 20 μL, which was incubated in 1.5 mL tubes at 37 °C for 24 h. The samples were deposited as 5 μL aliquots onto 200 mesh copper grids for 5 min, dried with filter paper, and then washed with ultra-pure water. After washing, the samples were negatively stained twice with 1% phosphotungstic acid for 5 min each time and then washed with ultra-pure water after each negative staining. The samples were dried at room temperature. Samples were examined under a low-voltage electron microscope 5 (Delong, Montreal, QC, Canada) at 5–6 kV.

### 4.5. Measurement of hSAA and Aβ Aggregation Inhibitory Activity of RA Using the MSHTS System

Various concentrations of RA were mixed with 40 μM hSAA or 25 μM Aβ and 30 nM QD_605_ in PBS. A total of 5 μL of sample was transferred to a 1536-well plate, where the plate was centrifuged at 3700 rpm for 5 min, and then incubated in a thermostatic incubator at 37 °C for 24 h. After incubation for 24 h, the inhibition of hSAA or Aβ aggregation by RA was observed under an inverted fluorescence microscope. The SD value of the image was calculated using the method described in Section 4.2.

## 5. Conclusions

We succeeded in the real-time imaging of the hSAA aggregation process using QDs and verified the hSAA aggregation process using ThT analysis. We evaluated the inhibitory activity of RA on hSAA aggregation using a novel MSHTS method based on nonspecific QD conjugation. In summary, our results provide further evidence of the accuracy and applicability of our independently developed MSHTS system for detecting amyloid aggregation and screening inhibitors.

## Figures and Tables

**Figure 1 ijms-25-11128-f001:**
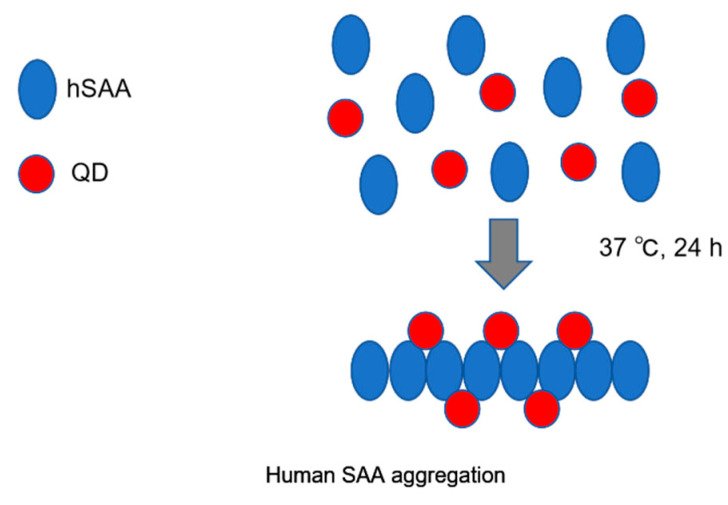
A schematic image of the visualization of hSAA aggregation using QDs. Initially, hSAA and unlabeled QD monomers were evenly distributed. To induce aggregation, hSAA and unlabeled QDs were incubated at 37 °C for 24 h. After 24 h of incubation, hSAA nonspecifically bound to unlabeled QDs during the aggregation process.

**Figure 2 ijms-25-11128-f002:**
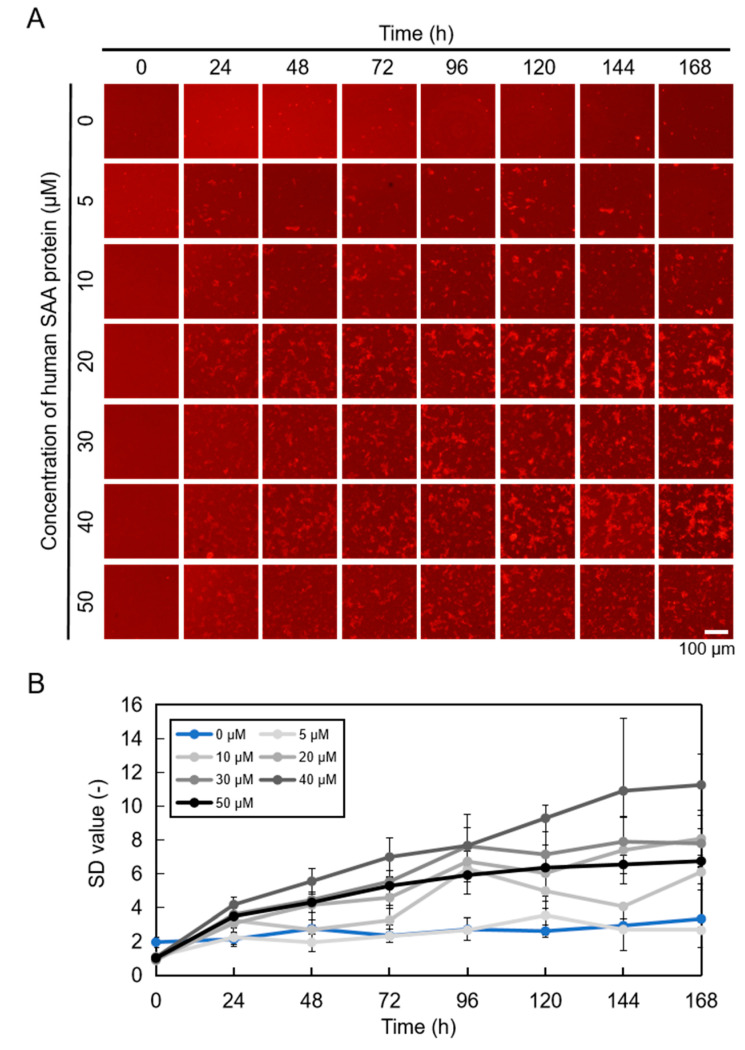
Imaging hSAA aggregation at different concentrations. (**A**) QDs were incubated with hSAA at different concentrations at 37 °C for 1 week, and images were captured using a fluorescence microscope every 24 h. (**B**) SD values of different concentrations of hSAA over time were determined using ImageJ software. Data represent mean values, and error bars indicate SDs derived from three separate experiments.

**Figure 3 ijms-25-11128-f003:**
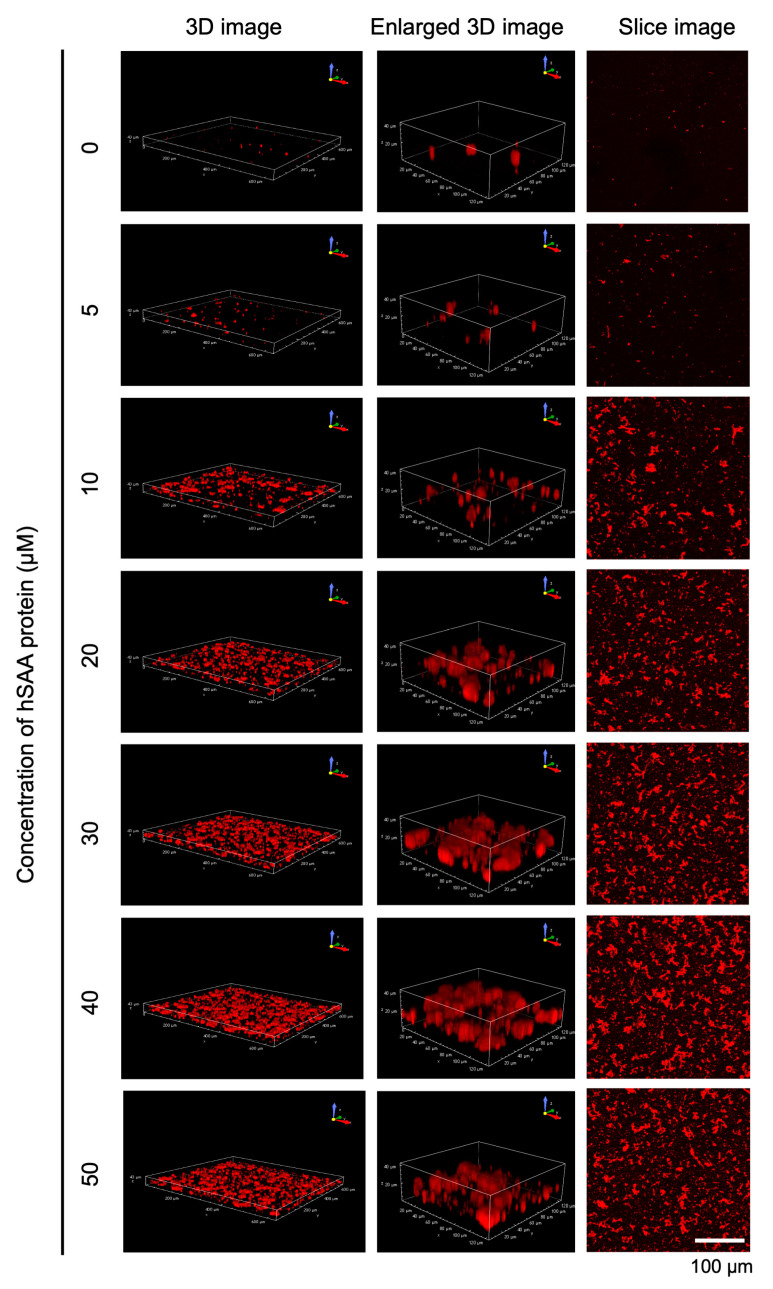
Left panel: Three-dimensional images of hSAA aggregation at different concentrations after incubation for 168 h. Middle panel: Cut and enlargement of the 3D images of the same part of the left panel. Right panel) Max intensity projection images from 3D images of hSAA aggregation.

**Figure 4 ijms-25-11128-f004:**
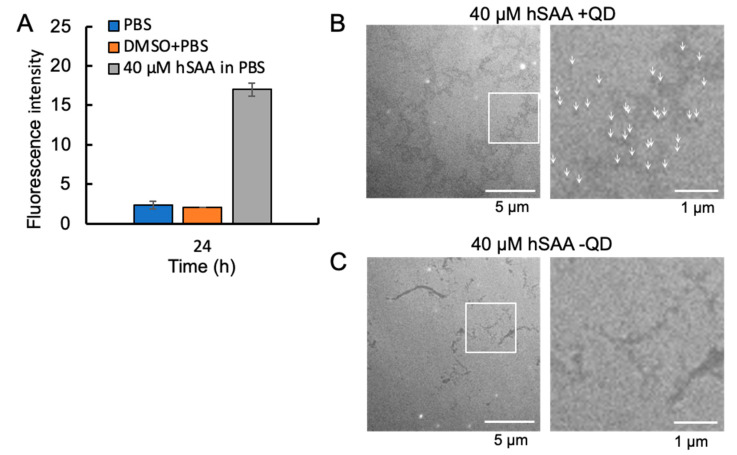
ThT assessment and TEM observation of 40 μM hSAA protein aggregation. (**A**) Fluorescence intensity of each group at 24 h. Data represent mean values, and error bars indicate SDs derived from three separate experiments. (**B**) Left panel: TEM observation of 40 µM hSAA protein fibrils (with QDs). Right panel: Cut and enlargement of the left panel. The white arrows indicate the QDs. (**C**) Left panel: TEM observation of 40 µM hSAA protein fibrils (without QDs). Right panel: Cut and enlargement of the left panel.

**Figure 5 ijms-25-11128-f005:**
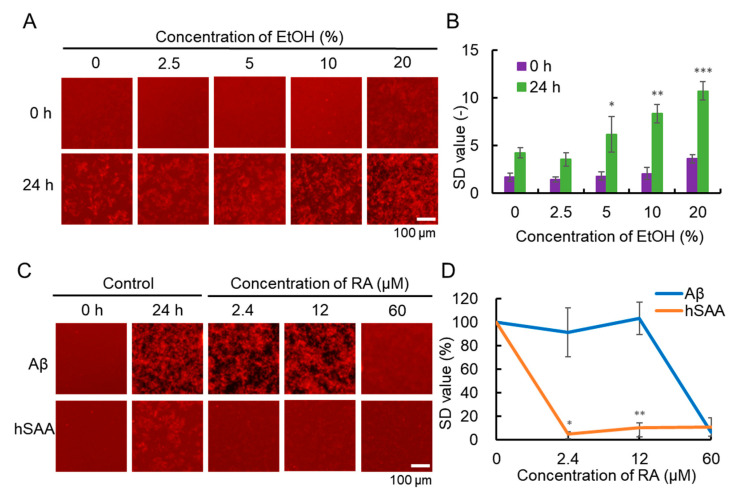
Effect of EtOH concentration on hSAA aggregation and inhibitory effect of RA on hSAA and amyloid β (Aβ) aggregation. (**A**) Fluorescence images of 40 μM hSAA after treatment with different concentrations of EtOH. (**B**) The SD values of the fluorescence images of hSAA aggregation at different concentrations of EtOH were measured for 24 h. Data represent mean values and error bars indicate SDs derived from three separate experiments. (**C**) Fluorescence images of 40 μM hSAA (5% EtOH, 9.6% DMSO in PBS) and 25 μM Aβ (5% EtOH, 2.5% DMSO in PBS) treated with different concentrations of RA. (**D**) Aggregation inhibition activity of RA on hSAA and Aβ. The SD values of fluorescence images of hSAA and Aβ aggregation were measured after 24 h. At 0%: The SD values of the fluorescence images of hSAA and Aβ of the control groups at 0 h in Figure 5C; 100%: The SD values of the fluorescence images of hSAA and Aβ of the control groups at 24 h in Figure 5C. Data represent mean values, and error bars indicate SDs derived from three separate experiments. *, **, and *** denote 0.01 < *p* < 0.05, 0.001< *p* < 0.01, and *p* < 0.001, respectively, which were determined by Welch’s *t*-test.

## Data Availability

The original contributions presented in the study are included in the article/Appendix A, further inquiries can be directed to the corresponding author/s.

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
