# Peer review of "Real-Time 3D Imaging and Inhibition Analysis of Human Serum Amyloid A Aggregations Using Quantum Dots"

_ijms, 2024, doi:10.3390/ijms252011128_

Round 1

Reviewer 1 Report

Comments and Suggestions for Authors

This manuscript describe a QDs based approach to visualize protein aggregation and screen inhibitors that block the course. The topic is interesting and of significance. However, several issues need to be well addressed or explained. Otherwise, we have to recommend to either reject or resubmit this manuscript.

1. This method has been published previously as shown in the references, therefore it is not so novel anymore.

2. The main novelty of the manuscript lies in human SAA was used this time, which is not novel enough as compared to mouse SAA studied by the authors before this project.

3. The inhibitor used in this study is rosmarinic acid (RA), which is a documented inhibitor. If authors discover a new inhibitor by using this system, then it can be a highlight.

4. If authors improve this system technically, that can also be encouraged.  

Comments on the Quality of English Language

Minor revision of the language might be needed. 

Author Response

Comments and Suggestions for Authors:

This manuscript describe a QDs based approach to visualize protein aggregation and screen inhibitors that block the course. The topic is interesting and of significance. However, several issues need to be well addressed or explained. Otherwise, we have to recommend to either reject or resubmit this manuscript.

Response:

Thank you for appreciating the interest and significance of our manuscript. We have carefully considered all your comments and made the necessary revisions. The responses to each comment are shown below.

Comment 1,2:

  1. This method has been published previously as shown in the references, therefore it is not so novel anymore.
  2. The main novelty of the manuscript lies in human SAA was used this time, which is not novel enough as compared to mouse SAA studied by the authors before this project.

Response 1, 2:

As the reviewer pointed out, the overall idea of this paper is not novel enough. However, amyloidosis caused by human SAA (hSAA) aggregation also leads to complications of other diseases, such as chronic inflammation in humans, so imaging aggregation and screening for its inhibitors is of great practical importance. In this paper, we have for the first time succeeded in analyzing the aggregation of hSAA over time in 3D using QDs and demonstrated that rosmarinic acid (RA) is a potent inhibitor of aggregation. We believe this research will hold great promise for the understanding of human amyloidosis and the discovery of preventive and therapeutic drugs.

In the revised manuscript, we added a comparison of the aggregation properties of hSAA and mSAA or Aβ (in lines 222 – 226 and 239 – 234). These findings demonstrate the applicability of this method to assess differences in the aggregation and/or inhibition properties of amyloidogenic proteins across species.

Comment 3:

  1. The inhibitor used in this study is rosmarinic acid (RA), which is a documented inhibitor. If authors discover a new inhibitor by using this system, then it can be a highlight.

Response 3:

As the reviewer comments, RA has been reported to inhibit the aggregation of various amyloidogenic proteins. However, the extent to which RA inhibits hSAA aggregation has remained unclear. In this paper, we compared the inhibitory effects of RA on the aggregation of Aβ and hSAA and demonstrated that RA inhibits the aggregation of hSAA more strongly than Aβ. In the revised manuscript, to compare the differences in the effects of RA on Aβ and hSAA aggregation, the changes in the SD values on the vertical axis of Figure 5D were normalized to percentages (in lines 172 – 178) according to reviewer#2 comment.

Comment 4:

  1. If authors improve this system technically, that can also be encouraged.

Response 4:

We described the advantages of this system for the evaluation of amyloid aggregation and inhibitors (in lines 272 – 276) according to the reviewer's suggestion.

Reviewer 2 Report

Comments and Suggestions for Authors

The manuscript, titled “Real-time 3D Imaging and Inhibition Analysis of Human Serum Amyloid A Aggregations Using Quantum Dots,” presents a study on monitoring hSAA aggregation in PBS using QDs and investigating the inhibitory effect of RA. This research is comparable to the same research group’s previous reports on mSAA or Aβ.

Two primary questions require clarification:

  1. QD Labeling and Aggregation: Previously, the amyloid aggregation was measured using QD-labeled amyloid. It is important to address the potential effect of QD labeling on aggregation and whether QDs themselves have a tendency to aggregate or stimulate amyloid aggreagation. Lines 188-197 should clarify the cytotoxicity of QDs versus non-specific binding characteristics. Additionally, an explanation for the non-toxicity of carbon QDs is needed. In Figure 4B, the QDs is not visual; further clarification is required. Figure 4C raises the question of whether hSAA forms aggregates in the absence of QDs.
  2. EtOH Effect and Aggregation Rate: The comparison of hSAA aggregation rates with Aβ is not entirely accurate because Aβ aggregation was measured using QDAβ(1-42)–Aβ(1-42) under different conditions, including 5% EtOH and 3% DMSO (ref 38). The effect of EtOH on aggregation needs to be carefully normalized in the inhibitor investigation. Figures 5C and 5D should clarify whether all conditions were equilibrated with the same EtOH concentration. To accurately compare the inhibitory effect of RA, the change in SD value may be more informative than the absolute SD value, as hSAA aggregation at 24h is significantly weaker than Aβ.

Author Response

Comments and Suggestions for Authors:

The manuscript, titled “Real-time 3D Imaging and Inhibition Analysis of Human Serum Amyloid A Aggregations Using Quantum Dots,” presents a study on monitoring hSAA aggregation in PBS using QDs and investigating the inhibitory effect of RA. This research is comparable to the same research group’s previous reports on mSAA or Aβ. Two primary questions require clarification:

Response: Thank you for your comments on our manuscript. We have carefully considered two primary questions and made the necessary revisions. The responses to each comment are shown below.

Comment 1:

  1. QD Labeling and Aggregation: Previously, the amyloid aggregation was measured using QD-labeled amyloid. It is important to address the potential effect of QD labeling on aggregation and whether QDs themselves have a tendency to aggregate or stimulate amyloid aggreagation. Lines 188-197 should clarify the cytotoxicity of QDs versus non-specific binding characteristics. Additionally, an explanation for the non-toxicity of carbon QDs is needed. In Figure 4B, the QDs is not visual; further clarification is required. Figure 4C raises the question of whether hSAA forms aggregates in the absence of QDs.

Response 1:

We referred to the results of the study in reference 5 to explain the potential effect of QD labeling on aggregation and whether QDs themselves have a tendency to aggregate or stimulate amyloid aggregation (in lines 197 - 202). We elaborated on the non-specific binding characteristics of QDs (in lines 202 – 204) and the non-toxicity of carbon QDs (in lines 212 – 214). To clearly show QDs, we modified Figure 4B and enlarged the left image of Figure 4B by 150×150 pixels to mark the QDs. We modified Figure 4C and clarified that the hSAA aggregation is not associated with QDs (in lines 137 – 140 and 146 – 147).

Comment 2:

  1. EtOH Effect and Aggregation Rate: The comparison of hSAA aggregation rates with Aβ is not entirely accurate because Aβ aggregation was measured using QDAβ(1-42)–Aβ(1-42) under different conditions, including 5% EtOH and 3% DMSO (ref 38). The effect of EtOH on aggregation needs to be carefully normalized in the inhibitor investigation. Figures 5C and 5D should clarify whether all conditions were equilibrated with the same EtOH concentration. To accurately compare the inhibitory effect of RA, the change in SD value may be more informative than the absolute SD value, as hSAA aggregation at 24h is significantly weaker than Aβ.

Response 2:

We have modified and explained the EtOH and DMSO concentrations of Figures 5C and 5D (in lines 158 – 161). To compare the changes in SD values of hSAA and Aβ, the SD values on the vertical axis in Figure 5D were normalized to percentages and explained in the figure legend (in lines 174 – 181).

Round 2

Reviewer 1 Report

Comments and Suggestions for Authors

This manuscript has been improved after revision. 

Reviewer 2 Report

Comments and Suggestions for Authors

The revised manuscript addresses all of my comments clearly.